# Recent Developments and Current Applications of Hydrogels in Osteoarthritis

**DOI:** 10.3390/bioengineering9040132

**Published:** 2022-03-24

**Authors:** Tianhao Zhao, Zhanqi Wei, Wei Zhu, Xisheng Weng

**Affiliations:** 1Department of Orthopaedics, Peking Union Medical College Hospital, Chinese Academy of Medical Sciences and Peking Union Medical College, Beijing 100730, China; zhaoth18@student.pumc.edu.cn (T.Z.); wei-zq15@mails.tsinghua.edu.cn (Z.W.); zhuwei9508@163.com (W.Z.); 2School of Medicine, Tsinghua University, Beijing 100084, China; 3Department of State Key Laboratory of Complex Severe and Rare Diseases, Peking Union Medical College Hospital, Chinese Academy of Medical Science and Peking Union Medical College, Beijing 100730, China

**Keywords:** osteoarthritis, hydrogels, hyaluronic acid, drug delivery system, mesenchymal stromal cells, entrapment, cartilage regeneration

## Abstract

Osteoarthritis (OA) is a common degenerative joint disease that causes disability if left untreated. The treatment of OA currently requires a proper delivery system that avoids the loss of therapeutic ingredients. Hydrogels are widely used in tissue engineering as a platform for carrying drugs and stem cells, and the anatomical environment of the limited joint cavity is suitable for hydrogel therapy. This review begins with a brief introduction to OA and hydrogels and illustrates the effects, including the analgesic effects, of hydrogel viscosupplementation on OA. Then, considering recent studies of hydrogels and OA, three main aspects, including drug delivery systems, mesenchymal stem cell entrapment, and cartilage regeneration, are described. Hydrogel delivery improves drug retention in the joint cavity, making it possible to deliver some drugs that are not suitable for traditional injection; hydrogels with characteristics similar to those of the extracellular matrix facilitate cell loading, proliferation, and migration; hydrogels can promote bone regeneration, depending on their own biochemical properties or on loaded proregenerative factors. These applications are interlinked and are often researched together.

## 1. Introduction

Osteoarthritis (OA) is a degenerative joint disease with symptoms of joint pain, cartilage degeneration, and osseous overgrowth [1]. Exercise, weight loss, nonsteroidal anti-inflammatory drug (NSAID) therapy, and intra-articular therapy are currently the main treatments for OA [2]. However, new strategies are needed to block the progression of OA and reduce the incidence of OA in the global population [3].

Hydrogels are formed by the polymerization of small molecules with a large number of hydrophilic groups that absorb plenty of water while forming crosslinks in solution [4]. Due to their composite structure rich in moisture and the crosslinking of polymers long chains, hydrogels exhibit certain elastic, adhesive, and mechanical properties that render them suitable as biomaterials [5], especially for applications in joint cavities, which are a small and relatively isolated environment [6].

At present, hydrogel treatments for OA have been widely studied to promote drug delivery and tissue regeneration and have proven to be effective, which has attracted increasing attention. This article will first introduce recent research on OA and hydrogels and then present the properties and functions of hydrogels implanted in joints. We will highlight four aspects of the function of hydrogels, i.e., their therapeutic effect, drug delivery applications, and cell entrapment and bone regeneration properties, with the aim of inspiring researchers to generate innovative ideas for the treatment of OA based on hydrogels.

## 2. OA

Joint cartilage is easily damaged during exercise, and this damage cannot be reversed due to the lack of blood vessels in the cartilage tissue and its low metabolic activity [1]. Long-term damage to joint cartilage eventually leads to OA and is accompanied by pain, inflammation, and loss of cartilage function; the inflammation usually invades all other joint structures, triggering osteophyte formation [7] and synovitis [8]. Based on Global Burden of Disease (GBD) data, OA showed an annual global increase of 0.32% in the age-standardized incidence rate (ASIR) or an approximately 9% increase from 1970 to 2017. Without age standardization, the data clearly show that population aging is accelerating the growth in the number of new cases of OA, with an increase in the crude incidence rate of approximately 102% from 1970 to 2017 [9]. The growing number of patients with severe and poorly managed OA often develop disabilities that put enormous financial pressure on individuals and healthcare systems [10].

Age, sex, obesity, etc., are all risk factors for joint damage and OA [11]; among them, age and obesity are major risk factors. Aging is accompanied by cumulative wear and degeneration, such as cartilage thinning and muscle atrophy, which eventually reduce the ability of joints to withstand undesirable conditions [12,13]. One meta-analysis concluded that overweight or obesity caused 24.6% of knee pain attacks in the examined cohort and that weight loss was effective in preventing OA [14]. Due to population aging and the prevalence of obesity, the number of people with symptomatic OA is likely to increase, posing a major challenge to public health systems [12].

OA is treated mainly through physical and pharmaceutical therapy [15]. Physical therapy includes weight loss and a proper exercise regimen. Weight loss can reduce mechanical loading and relieve injury, while proper exercise can slow OA progression by strengthening the muscles. Currently, commonly administered medications, such as corticosteroids and NSAIDs, focus on reducing pain and fighting inflammation. However, on the one hand, the long-term use of these drugs causes cardiovascular, digestive, and other side effects [16,17], while on the other hand, these drugs reach the joint cavity a long time after their oral administration or intravenous injection, achieving a low concentration in synovial cavities [18]. In addition, the current treatments are limited to relieving symptoms of the condition, and there are no drugs that facilitate OA patient recovery [19]. Therefore, further research is urgently needed to find a suitable treatment for blocking the progression of OA.

## 3. Hydrogels as a Biomaterial

Hydrogels with different three-dimensional structures, porosities, elasticities, and mechanical strengths can be produced by selecting different molecular monomer reagents and crosslinking them in a physical or chemical manner in aqueous solution to form insoluble network structures [20]. As biological materials, hydrogels maintain a high moisture content through a crosslinked network with complex physical and chemical properties, which can be used to build a microenvironment similar to that of the extracellular matrix and suitable for cell survival [21].

According to the source of the monomer, hydrogels are mainly divided into two categories: naturally forming hydrogels (hyaluronic acid (HA), collagen, fibrin (FB), alginate, etc.) and synthetic hydrogels (e.g., polyethylene glycol (PEG)-based hydrogels) [13]. Hyaluronic acid is suitable as a drug carrier due to its bio-viscoelasticity, biodegradability, non-immunogenicity, and biomedical benefits [22]. Collagen hydrogels have excellent biocompatibility, but their use is limited by their weak mechanical properties. A study tried to introduce cellulose nanofibers to collagen hydrogels to modulate their physical properties [23]. Regarding alginate hydrogels, some studies altered their molecular weight and introduced backbone chemical modifications and covalent crosslinking to improve their degradability, mechanical properties, and cell adhesion, so to adapt them for special applications of tissue engineering [24]. For example, oxidized alginate-based hydrogels have improved degradability and reactive groups [25]. PEG-based hydrogels are used after crosslinking [26] or modification (with the –SH group) [27] as simple scaffolds, and afford fast and gentle coagulation and excellent drug dispersion, respectively. Interestingly, synthetic hydrogels based on PEG were used as biomaterial earlier than natural hydrogels, probably due to the rapid development and maturation of polymeric technology and materials in the industry [28]. HA-based natural hydrogels have long been used as soft-tissue fillers or for viscosupplementation in the synovial cavity of patients, which are very different applications from those in the biomaterials field discussed herein [29]. Natural hydrogels were later gradually studied as biomaterials rather than drugs because of their excellent biocompatibility and degradability [30].

Hydrogels are now widely used to promote angiogenesis [31], tissue regeneration in the central nervous system [32], and cartilage generation in plastic surgery [33]. Both vascular tissue and cartilage need to bear stress and maintain elasticity, which are properties of hydrogels. In addition, porous hydrogels can be loaded with materials as small as drugs and as large as cells to serve a variety of functions for tissue regeneration. These features are also useful for OA treatment. Some of the hydrogels applied to OA are schematically presented in Appendix A and will be explained in the following sections.

## 4. Therapeutic Effects of Hydrogels

Regardless of other functions, hydrogels, especially HA-based ones, have a therapeutic effect on OA by themselves. Intra-articular injections of HA and chondroitin sulfate replenish joint fluid and reduce friction between joint cartilage surfaces to relieve pain [19]. A recent retrospective study concluded that intra-articular injections of HA in patients with mild and severe hip OA might relieve pain and improve function. Furthermore, three consecutive injections resulted in a better analgesic effect, which is the main effect of HA [34]. As shown in Figure 1, to improve both the retention rate of HA in the joint cavity and its efficacy, HA can be coupled with a thermosensitive polymer and injected into the joint cavity, while maintaining biocompatibility due to its reduced sensitivity to enzymes [35].

It should be noted that the efficacy of HA is limited. One analysis found that while both corticosteroids and HA reduced the risk of surgery for patients within 10 years, the risk of surgery and the cost of treatment were slightly higher in the HA cohort than in the corticosteroid cohort [36]. Another clinical trial found that corticosteroids and HA were able to alleviate OA progression, but patients treated with HA injections were at a higher risk of total knee replacement [37]. The possible combination of HA with other drugs is a more appropriate clinical approach. The results of a double-blind randomized experiment showed that long-term combined injections of HA and corticosteroids were more effective at relieving joint pain and improving motor function and physical condition than injections of HA alone [38]. Clinically, a Russian study recommends oral NSAIDs for patients with persistent symptoms of OA, supplemented by intra-articular HA and corticosteroids, especially if other drugs do not elicit a response [39].

In addition, hydrogels of nonmammalian origin, such as hydrogels based on chitosan and alginate, are also being examined for their therapeutic significance, as their thixotropy, nontoxicity, and drug release capability suggest their potential for viscosupplementation [40,41,42].

## 5. Hydrogel Implantation in Joints

As mentioned earlier, both monomeric (non-crosslinked) and polymeric (crosslinked) forms of HA have been accepted for the treatment of pain associated with knee OA [43]. It was later discovered that hydrogels with porous structures are able to release drugs slowly into the synovial cavity, promoting cell proliferation and tissue formation and thereby inhibiting inflammation and repairing cartilage [44]; these hydrogels could thus be used to develop sustained-release systems [30]. A cyclodextrin pseudopolyrotaxane system mixed with attapulgite was used to form a supramolecular hydrogel with a composite structure similar to that of “reinforced concrete”, allowing the sustained release and subsequent anti-inflammatory effects of diclofenac sodium [45]. Furthermore, the hydrogel backbone can also carry cells and deliver physicochemical signals and nutrients for cell growth [44,46]. As shown in Figure 2, a progenitor cell population and insulin-like growth factor-1 (GF-1) can be delivered simultaneously by thiolated gelatin/poly(ethylene glycol) diacrylate (PEGDA) interpenetrating network (IPN) hydrogels, and loading GF-1 in coacervates mixed with the hydrogel can ensure a long-term sustained effect on stem cells in terms of proliferation and tissue regeneration [47].

Biocompatibility, biofunctionality, mechanical properties, and adjustable degradation of polymer hydrogels are basic characteristics of hydrogels used intra-articularly [48]. Naturally forming hydrogels have outstanding biocompatibility, low immunogenicity, low cytotoxicity, and an excellent capability to promote cell adhesion and proliferation and new tissue regeneration compared to synthetic hydrogels [48,49]. However, natural hydrogels are degraded via diverse pathways in vivo, which reduces their efficiency. Physiologically, HA is degraded by intracellular and serum enzymes or decomposed by heat and oxidants [43]. FB is rapidly broken down by plasmin. By adding fibrinolytic inhibitors [50] or inducing polymerization with synthetic hydrogel monomers such as PEG [51], the degradation rate of natural hydrogels can be tuned, and their stability can be improved. The variety of natural hydrogels is limited; thus, natural hydrogels sometimes cannot meet applications’ needs. Synthetic hydrogels are of interest for achieving longer efficacy, a higher gel strength, and customizable functionality and degradability, although their poor compatibility still has to be overcome [52]. Some researchers have tried to develop composite hydrogels using complementary natural and synthetic sources, such as a double-network hydrogel of poloxamer–heparin/gellan gum [53] and a hydrogel platform based on PEG and gelatin [54]. The poloxamer–heparin/gellan gum hydrogel formed a microenvironment conducive to stem cell growth, and in vivo experiments showed that it supports bone marrow stem cell survival, attachment, and extracellular matrix production. The PEG/gelatin hydrogel could effectively promote cell differentiation, with an effect better than that of the heterogeneous protein mixture Matrigel and exhibited improved strength due to the covalent binding of PEG.

## 6. Hydrogel-Based Intra-Articular Drug Delivery

As mentioned earlier, the intra-articular injection of HA polymers into patients with OA can relieve pain. It is known that small-molecule NSAIDs often undergo rapid depletion. Experiments have shown that some drugs (paracetamol) and proteins (albumin) are not retained long enough in the joint cavity and may not be suitable for injection therapy in free form [55]. Therefore, some kind of matrix is needed to carry drugs and release them locally in a sustained manner so to achieve a long-term treatment. Hydrogels can maintain high local concentrations of drugs for a long time. Abundant crosslinking and expansion upon water absorption allow hydrogels to form a loose and porous microenvironment, which can be loaded with a variety of drugs. The ratio of feedstock or synthesis can be changed to adjust the size and density of the voids and adapt hydrogels to the molecular size of drugs and to the required rate of drug diffusion [56]. A biodegradable ternary hydrogel from oxidized dextran (Dex-ox), gelatin, and HA was injected to deliver two different anti-inflammatory drugs, i.e., naproxen (NSAID) and dexamethasone (Dex). New Zealand rabbits in the experimental group presented a low macroscopic degree of OA in the injected knees and better recovery [57]. Tyramine-modified HA (HA-Tyr) hydrogels encapsulating Dex resulted in the successful treatment of rheumatoid arthritis (RA) [58]. Although RA and OA are not the same disease, HA-Tyr hydrogels still provide ideas for intra-articular corticosteroid delivery. Good therapeutic effects were also achieved by the delivery of anti-inflammatory drugs such as indomethacin and celecoxib using a hydrogel system in animal experiments [59,60,61]. Through the use of combinations of hydrogels and other biomaterials, it is possible to prolong the sustained release of drugs and improve the performance of hydrogels. An injectable carboxymethyl chitosan–methylcellulose–pluronic hydrogel encapsulating meloxicam-loaded nanoparticles showed a reduced rate of gel degradation. Meloxicam was released separately from the gel and the nanoparticles, which extended the delivery time relative to the use of the hydrogel alone [62].

Hydrogels have also been used in an attempt to sustain the concentration and functioning of some drugs that are unstable in solution, such as kartogenin (KGN), in the joint [63]. KGN stimulates the differentiation of multipotent mesenchymal stem cells (MSCs) and the subsequent repair of damaged cartilage. A carrier system based on halloysite nanotubes and a laponite hydrogel demonstrated the slow release of KGN over 7 days [64], while another PEG-HA hydrogel reduced the release rate of KGN via covalent integration [65]. Cordycepin has been shown to inhibit the expression of ADAMTS-5 and MMP13 in IL-1β-induced OA, thus preventing inflammation. A hyaluronic acid methacrylate (HAMA) hydrogel together with chitosan microspheres could support the long-term release of cordycepin in a controlled manner [66]. Another ADAMTS-5 inhibitor (114810), with an HA hydrogel as a carrier, promoted cartilage healing in an osteochondral defect model [67]. Furthermore, cordycepin protected chondrocytes by facilitating autophagy. Several autophagy activators, including sinomenium and rapamycin, can also be delivered to ameliorate cartilage matrix degradation [68,69].

In addition to small-molecule drugs, some protein drugs, nucleic acids, and tissue extracts can be delivered using modified hydrogels. The affinity of HA itself for proteins is not sufficiently high. Sulfated HA showed not only improved protein sequestration but also greater resistance to hyaluronidase-induced decomposition, allowing the long-term action of the protein-hydrogel delivery system [70]. Platelet-rich plasma (PRP) promotes cartilage matrix synthesis and cartilage repair because it contains growth factors. However, the injected dose of PRP is easily lost from the synovial cavity, and the effect is very fast and unsustainable [71]. Gelatin (GLT) hydrogel microspheres loaded with PRP achieved increased expression of proteoglycan core protein mRNA in articular cartilage [72], while a genipin (GP)-HA/fucoidan (FD)/gelatin system facilitated the sustained release of PRP growth factors [73]. Exosomes are widely present in body fluids, are filled with proteins and RNA [74], can be used to encapsulate drugs, and can be delivered by hydrogels. Some researchers incorporated PRP-derived exosomes in an hydrogel matrix consisting of an optimal mixture of poloxamer-407 and poloxamer-188, significantly increasing the lifespan and retention of exosomes in joints and thus the duration of PRP release, which lasted continuously for 28 days, showing therapeutic effects [75].

Some responsive hydrogels are used to ensure that drugs can be evenly and stably dispersed before implantation, ensuring sustained release; the most common of these are temperature-responsive hydrogels. As an inexpensive synthetic corticosteroid, Dex is often used in the development of temperature-responsive hydrogels. The Dex-loaded thermosensitive hydrogel developed by Qi-Shan Wang et al. coagulated at 37 °C, and the cumulative release curve after coagulation showed that Dex was released slowly over 7 days, resulting in an analgesic effect and inflammatory factors downregulation in mouse OA models [76]. The formation of the thermal response of another N-(2-hydroxypropyl) methacrylamide (HPMA) copolymer-based Dex prodrug was accidental, and the researchers attributed this effect to the high level of Dex precursor in the polymer solutions. This precursor drug solution gelled in the joint or at temperatures above 30 °C and was retained for 1 month, during which the released precursor drug could be processed by phagocytes to produce free Dex, improving symptoms of OA [77]. A celecoxib-loaded hydrogel based on a fully acetyl-capped ε-caprolactone-co-lactide (PCLA)–PEG–PCLA triblock copolymer transitioned to a gel at 37 °C and showed sustained celecoxib release for 90 days after a 10-day lag period [59]. Glucosamine (GlcN)-loaded thermosensitive hydrogels based on poloxamer-407 and poloxamer-188 slowly released GlcN in vitro and decreased the degree of swelling and the levels of inflammatory factors after intra-articular administration to treat OA in rabbits [78]. Temperature-responsive hydrogels also exhibit the therapeutic effects of the hydrogels themselves. As shown in Figure 3, a hyalomer containing poloxamer-407 (PX) as a thermogelling agent showed strong antinociceptive and antiedematous effects [79].

## 7. MSC Entrapment

MSCs are multipotent progenitor cells from bone marrow and adipose tissue that can be induced to differentiate into bone cells, fat cells, and chondrocytes. In OA treatment, MSCs are often used to promote the formation of hyaline-like persistent cartilage through implantation during surgery [80]. However, the application of MSCs still has limitations, because, for example, of the shear force generated by traditional injection methods which leads to massive cell death and MSC leakage from the joint cavity, causing poor targeting. Therefore, a convenient cell delivery system, such as hydrogels, is needed [81].

Early studies have already shown that MSC-HA products led to good cartilage regeneration during the follow-up period in clinical trials [82]. Properly designed, friction-resistant, mechanically strong hydrogel structures ensure cellular integrity. As shown in Figure 4, recent research has led to the development of a DNA supramolecular hydrogel that has extraordinary strength in not only resisting shear forces both in vitro and in vivo but also tolerating friction due to joint motion [83]. Another study based on a natural chondrogenic FB/HA hydrogel modified with methacrylic anhydride (MA) to enhance its mechanical properties also reported sufficient material properties to increase the potential of MSC proliferation and cartilage formation [84]. With the aim of improving cell retention at the treatment site, a gelatin-based three-dimensional particle gel was loaded with low doses of human umbilical cord-derived MSCs, enhancing cell survival time and cell retention while maintaining therapeutic effects similar to those of high-dose free MSCs [85]. The ability of hydrogels to retain MSCs depends not only on a solid-like gel morphology but also on modifications of the hydrogel and improvement of its adhesion capacity, which involves cell–hydrogel adhesion and surrounding cartilage tissue–hydrogel adhesion [86,87,88]. A photocrosslinkable PRP hydrogel glue developed using photoresponsive HA was transformed into a structure with tissue adhesion under light exposure, which was related to the production of aldehyde groups by the photoinduced imine crosslinking reaction [87].

The unsatisfactory differentiation ability of MSCs from different sources might be obviated by loading hydrogels with differentiation inducers. An alginate/HA hydrogel enabled the transfection of MSCs by plasmids that carried transforming growth factor-beta 3 (pTGF-β3) or bone morphogenetic protein 2 (pBMP2), which was not possible using naked pDNA. The delivery of TGF-β3 and BMP2 genes and their subsequent cell-mediated expression promoted the accumulation of cartilage matrix [89]. Osteoblasts, a type of non-MSC, also showed the ability to form bone tissue in mouse models through DNA delivery via alginate hydrogels [90]. The migration of MSCs influences cartilage repair. Stromal cell-derived factor 1 alpha (SDF-1α) plays a crucial role in the activation, mobilization, homing, and migration of MSCs. The use of a chitosan-based hydrogel embedded with SDF-1α affected the migration of MSCs in vitro and in vivo, remarkably promoting stem cell homing and cartilage repair in the OA model [91]. Interestingly, there is a view that MSCs have a low probability of targeted differentiation into chondrocytes in the pathological environment of OA. Transplanted MSCs can play a reparative role because of their secretory and immunomodulatory functions [81]. Using hydrogels as a delivery system for MSC-derived small extracellular vesicles (MSC-sEVs) rather than MSCs themselves proved that vesicles also have the ability to improve OA [92].

## 8. Articular Cartilage Regeneration

The use of hydrogels to repair degenerated cartilage in OA is based on two main concepts, one of which is the transplant of autologous cells and the stimulation the proliferative potential of the surrounding stem cells [49]. Autologous cells can be MSCs, as we mentioned in the MSC entrapment section, or fully differentiated chondrocytes [93]. Hydrogel-encapsulated chondrocytes can be induced to promote the regeneration of the surrounding cartilage tissue by secreting a collagen-rich extracellular matrix [94].

Another concept is to rely not on encapsulated cells but on the hydrogel itself to promote bone regeneration. Hydrogels mixed with cartilage matrix components such as collagen [86], chondroitin sulfate [95], and HA [96] tend to build a microenvironment that promotes the proliferation of the surrounding cartilage tissue and fill damaged cartilage even without encapsulating cells, thereby improving cartilage defects. Hyaluronan/p (HPMAm-lac)–PEG hydrogels slowly released HA during their degradation, resulting in bone remineralization and proteoglycan production in mouse OA models, while also leading to the downregulation of proinflammatory mediators (e.g., TNF-α and NF-κB) and proinflammatory cytokines [97]. A 3D-printed hydrogel functionalized with aggrecan that supported the cellular fraction of bone marrow demonstrated tremendous improvement in the regenerated cartilage tissue quality in a lapine model [98]. In the group treated with the aggrecan-functionalized scaffold, the growth of cartilage tissue and cell density increased significantly.

There have also been studies dedicated to the combination of additives and hydrogels. The addition of hydroxyapatite and bone morphogenetic protein may be beneficial to further enhance the role of hydrogels in cartilage repair [99]. An interpenetrating polymer network scaffold of sodium hyaluronate and sodium alginate combined with berberine could regenerate both cartilage and subchondral bone. The subchondral bone was partially repaired by activating the WNT signaling pathway, and the cartilage was protected from degeneration through the upregulation of autophagy [100]. Endogenous stem cells may also be able to participate in bone regeneration. As shown in Figure 5, a microRNA that targets miR-221 (antimiR-221) was delivered via a hydrogel to guide cartilage repair in situ by endogenous cells. AntimiR-221 blocked the expression of miR-221 and enhanced chondrogenesis in vitro after the transfection of human bone marrow-derived mesenchymal stromal cells. The FB/HA hydrogels strongly retained the functional antimiR-221 over 14 days of in vitro culture. Implanting FB/HA loaded with anti-miR-221/Lipofectamine into mouse cartilage defects significantly enhanced cartilage repair by endogenous cells [101]. A crosslinked network of alginate–dopamine, chondroitin sulfate, and regenerated silk fibroin (AD/CS/RSF) encapsulating exosomes recruited MSCs, promoted MSC proliferation and differentiation, and accelerated the in situ regeneration of cartilage defects, resulting in extracellular matrix remodeling in the patella groove of rats [102].

## 9. Conclusions and Future Perspectives

There are several different strategies for applying hydrogels in the treatment of OA, including strategies involving drug delivery, MSC entrapment, and bone (cartilage) regeneration. From the perspective of traditional therapies for reducing inflammation and relieving pain, the development of hydrogel delivery systems has solved the issue of the side effects caused by traditional systemic drug delivery routes and the rapid loss of drugs injected into joints to a certain extent. However, for the wide application of hydrogel-based treatments in the clinic, the maximum concentration and retention time of drugs in hydrogels need to be improved to achieve long-term efficacy and reduce the number of injections. After all, intra-articular injections are inherently invasive and can cause joint inflammation. The combination of studies on new drugs and hydrogels related to OA has allowed improving the retention rate of drugs in the lesion area, thus ensuring reliable efficacy. In terms of completely curing OA, strategies consisting of porous hydrogels combined with stem cells taking full advantage of the characteristics of suitable small-molecule drugs, organic matter, and cells for achieving the retention and sustained release of small molecules are promising. The microstructure of a well-designed hydrogel can adapt to the shape of stem cells and can promote stem cell proliferation, migration, and differentiation and even induce dedifferentiation. In terms of tissue regeneration, in addition to loading hydrogels with exogenous stem cells, there has also been research into the development of methods to induce endogenous cell proliferation. We believe that although the latter may be more challenging due to the rarity of endogenous stem cells, it can avoid the risk of an immune response. It should be noted that these various applications are not independent of each other. The properties that allow hydrogels to retain drugs also render hydrogels suitable for carrying differentiation-inducing molecules, bone regeneration signals, and nutrients; additionally, hydrogel-encapsulated stem cells are involved in tissue regeneration, suggesting the multifunctional involvement of hydrogels in such therapies. Thus, researchers need to think holistically about the role that biomaterials play in therapy.

The combination of hydrogels with other biomaterials is also gradually becoming a common strategy to improve clinical performance [68,103,104]. Although the current hydrogel strategies still present challenges, including the need to accurately deliver drugs to cartilage rather than to other tissues [105] and mechanical property deficiencies [106], these strategies are still generally very promising.

## Figures and Tables

**Figure 1 bioengineering-09-00132-f001:**
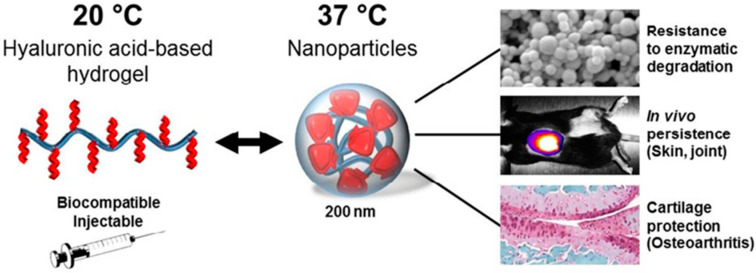
To overcome the inconvenience of repeated injections and the rapid degradation of exogenous HA, HA is conjugated to a thermosensitive polymer, enabling the spontaneous formation of nanoparticles (HA Nano) at body temperature, Reprinted from Ref. [30].

**Figure 2 bioengineering-09-00132-f002:**
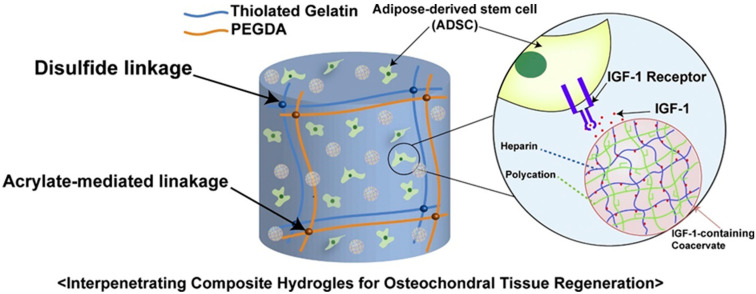
An implantable dual-delivery platform was developed using a tertiary complex of poly(ethylene argininylaspartate diglyceride) (PEAD) polycation, heparin, and cargo insulin-like GF-1 in a thiolated gelatin/PEGDA IPN hydrogel, Reprinted from Ref. [42].

**Figure 3 bioengineering-09-00132-f003:**
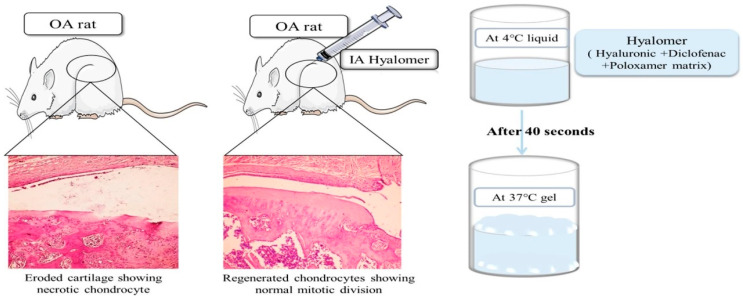
Hanafy et al. aimed to prepare a hyalomer for intra-articular (IA) injection and in situ gel formation. The hyalomer contained poloxamer-407 (PX) as a thermogelling agent, HA, and diclofenac potassium (DK) as an anti-inflammatory agent, Reprinted from Ref. [74].

**Figure 4 bioengineering-09-00132-f004:**
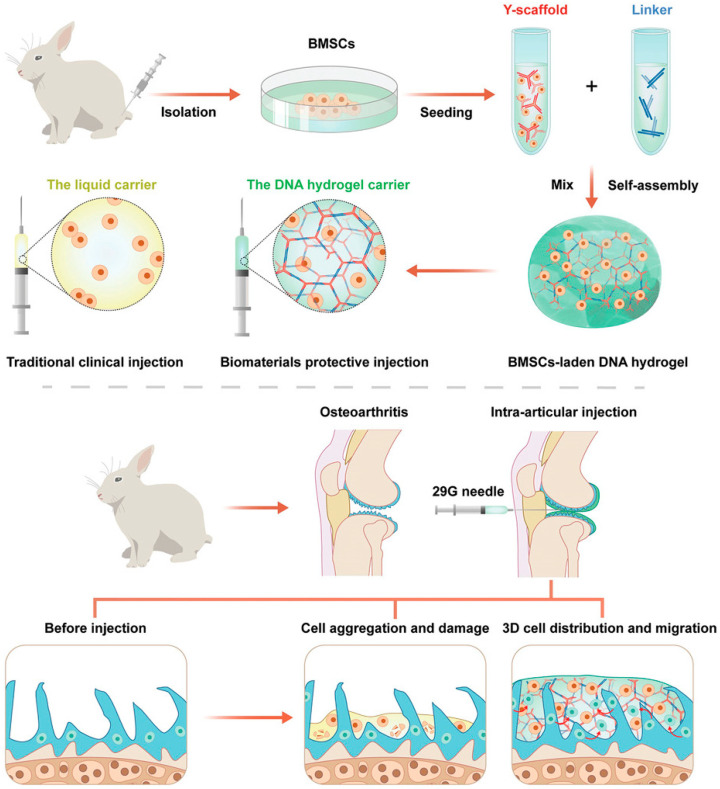
Schematic diagram of an anti-friction MSC delivery system, which showed an improved therapeutic effect on severe OA, Reprinted from Ref. [78].

**Figure 5 bioengineering-09-00132-f005:**
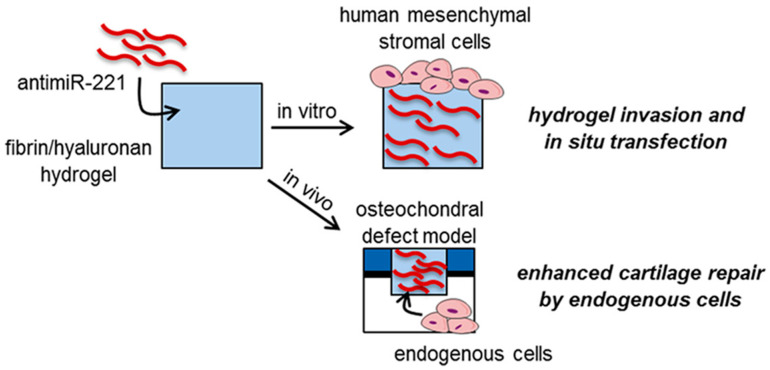
Lolli et al. investigated a new strategy based on hydrogel-mediated delivery of a locked nucleic acid microRNA inhibitor targeting miR-221 (antimiR-221) to guide in situ cartilage repair by endogenous cells, Reprinted from Ref. [96].

## Data Availability

Not applicable.

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
