# Peer review of "Recent Developments and Current Applications of Hydrogels in Osteoarthritis"

_bioengineering, 2022, doi:10.3390/bioengineering9040132_

Round 1

Reviewer 1 Report

The manuscript describes in detail the use of hydrogels in the treatment of osteoarthritis.
The chapters regarding the different literature studies such as the use in  drug delivery or stem cell entrapment or in articular cartilage regeneration, are comprehensive and well organised.  I Suggest not to use the term encapsulation, but to use terms such as loading or entrapment, reserving the term encapsulation for shell-core systems, which is not the case with hydrogels.

On the other hand, I think that the initial paragraphs should give a little more detail on the chemical and physical characteristics of the different types of hydrogels, with particular emphasis on thermo-reversible gels.
In addition, a table summarising the different applications described in the literature might be of help to the reader. 

Reviewer 2 Report

It is an informative review on the field of hydrogels use in orthopedics. I have detected some language errors that make understanding difficult to me and I marked these areas that need reduction with red. 

E.g. Line 74 "oral or intravenous injection" of course the authors do not mean oral injection but oral administration but this should be changed. 

Line 91 "synthetic hydrogels based on PEG have developed natural hydrogels" The reader wonders how a natural substance could be "developed" same line 96. Do you mean discovered? or used? or prepared?

line 161 "degradation is difficult to control artificially" I cannot understand what do you mean here. Degradation can be controlled by any other means ?

Line 167 " a longer service life" do you mean longer efficacy?

Reviewer 3 Report

The current manuscript provides a literature account of hydrogel interventions for osteoarthritis. I recommend some suggestions for improvement of the manuscript as follows:

1. The extent and coverage of biomaterials is very limited in the manuscript. Most of the studies mentioned are from Hyaluronic acid and PEG perspective. I feel more biomaterials could have been included and it will benefit to provide a table for the same.

2. The aspect of artificial meniscal fluid and its applications in achieving pain and physical relief are missing from the manuscript.

3. There are no original images in the article and all of them have been reproduced from previous studies.

4. Another part that is lacking in the study is the scaffold part. Most of the studies mentioned are as hydrogels but well known porous scaffolds and sponges are missing from the review.

Round 2

Reviewer 3 Report

I am satisfied with most of the rebuttals. However, I couldn't find the Table in the manuscript as mentioned by the authors: "We also added a summary table of hydrogels to show readers the parts that we have not explained in detail in the text after "Hydrogels as a Biomaterial"."